Evidence for a trophic cascade on rocky reefs following sea star mass mortality in British Columbia

Schultz Jessica A. 1 2 jessica.schultz@vanaqua.org
Cloutier Ryan N. 3
Côté Isabelle M. 1
1 Earth to Ocean Group, Department of Biological Sciences, Simon Fraser University , Burnaby, British Columbia , Canada
2 Coastal Ocean Research Institute, Vancouver Aquarium Marine Science Centre , Vancouver, British Columbia , Canada
3 Stantec Consulting Ltd. , Burnaby, British Columbia , Canada
Taylor Richard
Electronic publication date: 2016 Apr 26
Publication date: 2016
Volume: 4
Electronic Location ID: e1980
Received 2015 Nov 26; Accepted 2016 Apr 6
Copyright: ©2016 Schultz et al.
Copyright year: 2016
Copyright holder: Schultz et al.
License: This is an open access article distributed under the terms of the Creative Commons Attribution License, which permits unrestricted use, distribution, reproduction and adaptation in any medium and for any purpose provided that it is properly attributed. For attribution, the original author(s), title, publication source (PeerJ) and either DOI or URL of the article must be cited.
License URL: https://creativecommons.org/licenses/by/4.0/

Keywords: Marine diseases, Starfish, Community shifts, Mass mortality, Environmental change, Sea star wasting syndrome, Echinoderm population

Funding: Vancouver Aquarium Howe Sound Research Program Canadian Healthy Oceans Network Discovery grant of the Natural Sciences and Engineering Research Council of Canada Jessica Schultz received funding through the Vancouver Aquarium Howe Sound Research Program. Ryan N. Cloutier received funding through the Canadian Healthy Oceans Network. Isabelle M. Côté received funding through a Discovery grant of the Natural Sciences and Engineering Research Council of Canada. The funders had no role in study design, data collection and analysis, decision to publish, or preparation of the manuscript.

==============================
Echinoderm population collapses, driven by disease outbreaks and climatic events, may be important drivers of population dynamics, ecological shifts and biodiversity. The northeast Pacific recently experienced a mass mortality of sea stars. In Howe Sound, British Columbia, the sunflower star Pycnopodia helianthoides—a previously abundant predator of bottom-dwelling invertebrates—began to show signs of a wasting syndrome in early September 2013, and dense aggregations disappeared from many sites in a matter of weeks. Here, we assess changes in subtidal community composition by comparing the abundance of fish, invertebrates and macroalgae at 20 sites in Howe Sound before and after the 2013 sea star mortality to evaluate evidence for a trophic cascade. We observed changes in the abundance of several species after the sea star mortality, most notably a four-fold increase in the number of green sea urchins, Strongylocentrotus droebachiensis, and a significant decline in kelp cover, which are together consistent with a trophic cascade. Qualitative data on the abundance of sunflower stars and green urchins from a citizen science database show that the patterns of echinoderm abundance detected at our study sites reflected wider local trends. The trophic cascade evident at the scale of Howe Sound was observed at half of the study sites. It remains unclear whether the urchin response was triggered directly, via a reduction in urchin mortality, or indirectly, via a shift in urchin distribution into areas previously occupied by the predatory sea stars. Understanding the ecological implications of sudden and extreme population declines may further elucidate the role of echinoderms in temperate seas, and provide insight into the resilience of marine ecosystems to biological disturbances.

Introduction

Echinoderms can be subject to dramatic population fluctuations (Uthicke, Schaffelke & Byrne, 2009). Rapid declines are often driven by disease or extreme climatic events. For example, the spread of mass mortality of the black sea urchin, Diadema antillarum, in the 1980s suggests that it was most likely caused by a pathogen (Lessios, Robertson & Cubit, 1984). The event impacted an estimated 3.5 million km2 of the Caribbean region, causing up to 99% urchin mortality at some sites (Lessios, 1988). While the precipitous decline of Diadema was a unique occurrence, other echinoderm mass mortality events occur repeatedly. On the Atlantic coast of North America, an amoeboid parasite causes episodic mortality events in green sea urchins, Strongylocentrotus droebachiensis (Jones & Scheibling, 1985), which are linked to hurricanes and are predicted to increase in frequency with climate change (Scheibling & Lauzon-Guay, 2010). Similarly, recurring events of wasting disease involving asteroids (sea stars), echinoids (sea urchins) and holothurians (sea cucumbers) in the Channel Islands, California, are associated with climate regime shifts and extreme weather events (Engle, 1994; Eckert, Engle & Kushner, 2000).

Because sea stars and sea urchins play key ecological roles in many marine ecosystems, echinoderm population collapses can be important drivers of biodiversity, population dynamics and ecological shifts. In fact, the term ‘keystone predator’ was originally coined for the purple star, Pisaster ochraceus, after experiments showed that its absence led to significant decreases in intertidal biodiversity (Paine, 1966). Many other echinoderm species have since been shown to influence community composition through predation or herbivory. These effects are apparent on coral reefs following echinoderm population booms (e.g., coral cover declines owing to eruptive crown-of-thorns star, Acanthaster planci Sano et al., 1984) or busts (e.g., the transition from coral- to algae-dominated reefs following the D. antillarum mortality event Carpenter, 1990). On temperate rocky reefs, fluctuations in the abundance of herbivorous urchins can also result in major community shifts, from kelp forests to urchin barrens and back again (Estes & Duggins, 1995; Steneck et al., 2003).

The northeast Pacific region has recently experienced a protracted mass mortality of sea stars that might rival the magnitude of the Diadema die-off of the 1980s (Johnson, 2016). The event was first noticed on the Olympic coast of Washington in June 2013 (Hewson et al., 2014). In affected sea stars, the signs progress from a loss of turgor pressure, to lesions and ruptures of the body wall and autotomization of arms, and ultimately, disintegration and death (Fig. 1). The wasting syndrome has continued through 2014 and 2015, and has so far affected some 20 species from Alaska to Southern California (Stockstad, 2014). A virus may be involved (Hewson et al., 2014), but the precise causes and contributing factors remain poorly understood. Moreover, little is known so far of the extent and ecological consequences of this sea star mortality event at any location.

Figure 1 Progression of sea star wasting disease.

(A) A healthy-looking specimen of P. helianthoides moves across the kelp, Agarum fimbriatum. (B) Afflicted sea stars exhibit a loss of turgor pressure and body wall ruptures, followed by (C) limb autotomization, disintegration and death. Photos by Donna Gibbs.

Many of the affected sea stars were predatory species, raising the possibility of trophic cascades associated with their disappearance and marked community restructuring. In Howe Sound, southern Strait of Georgia, British Columbia, the sunflower star Pycnopodia helianthoides showed signs of advanced wasting in early September 2013. Dense aggregations disappeared from many sites in a matter of weeks (J Schultz, pers. obs., 2013). This species is one of the world’s largest predatory sea stars and it consumes a variety of prey, including echinoderms, gastropods and crustaceans (Herrlinger, 1983; Shivji et al., 1983). In areas that lack other predators such as sea otters Enhydra lutris, such as in Howe Sound, sunflower stars can become the dominant predator of urchins (Duggins, 1983). By altering the abundance and/or distribution of sea urchins, which in turn can have a conspicuous impact on the abundance of kelp, sunflower stars can influence the formation and persistence of urchin barrens, i.e., areas devoid of kelp because of the grazing activity of urchins (Duggins, 1981). Indeed, most well-substantiated examples of tri-trophic cascades in rocky subtidal ecosystems involve urchins as prey and major herbivore (Pinnegar et al., 2000). We therefore expected that Pycnopodia prey, in particular urchins, would increase in abundance following the disappearance of their major predator, leading to reductions in kelp cover.

Here, we evaluate the extent of mortality of P. helianthoides in Howe Sound and test whether changes in the benthic community following the rapid decline of this predatory sea star are consistent with the hypothesis of a trophic cascade. We compare rocky reef community composition before and after the mass mortality using quantitative data derived from subtidal transects and qualitative information gathered by citizen scientists. In doing so, we provide empirical evidence that a trophic cascade quickly followed what might be one of the largest wildlife die-off events ever recorded (Johnson, 2016).

Materials and Methods

Subtidal surveys

We compared sunflower star abundance and benthic community composition before (2009/2010) and after (2014) the 2013 wasting event using scuba-based surveys of 20 sites in Howe Sound, British Columbia (BC), Canada (Fig. 2). Surveys before the wasting event were conducted as part of a study of rockfish (Sebastes spp) habitat (Cloutier, 2011). We repeated these surveys after the wasting event using the same method, at the same GPS locations, depths (within 2 m) and time of year (within 14 days). Ten sites were surveyed in early summer (June–July) and 10 sites in late summer (August–October). In all surveys, we recorded the abundance of 18 taxa (species or species groups) of common benthic fishes and invertebrates (Table 1).

Figure 2 Rocky reef survey sites in Howe Sound, British Columbia.

Benthic community composition was assessed at each of the 20 sites once in 2009 or 2010 and again in 2014. A mass mortality of sea stars occurred in the summer and fall of 2013 in this area. A site-level trophic cascade following the mortality was detectable at some sites (solid circles) but not others (open circles). (Map data © 2015 WorldMap).

Table 1 Taxa recorded during subtidal surveys in Howe Sound, British Columbia.

Mean density and standard deviation per 15 m2 are given for each taxon as recorded before and after the sea star mortality event.

Taxon	Species or genera included in taxon	Mean density (SD)	
		Before	After	
Invertebrates	
Sunflower star	Pycnopodia helianthoides Brandt, 1835	6.4 (11.4)	0.9 (3.3)	
Green urchin	Strongylocentrotus droebachiensis O. F. Müller, 1776	18.3 (41.0)	77.2 (157.4)	
Red urchin	Strongylocentrotus franciscanus Aggasiz, 1863	0.4 (0.9)	0.3 (0.6)	
White urchin	Strongylocentrotus pallidus G. O. Sars, 1871	1.1 (2.0)	0.3 (0.4)	
California cucumber	Parastichopus californicus Linnaeus, 1758	6.1 (9.0)	13.1 (8.9)	
Dungeness crab	Metacarcinus magister Dana, 1852	0.1 (0.2)	0.0	
Red rock crab	Cancer productus Randall, 1839	0.1 (0.3)	0.4 (0.7)	
Spot prawn	Pandalus platyceros Brandt, 1851	22.1 (89.1)	0.3 (0.8)	
Squat lobster	Munida quadrispina Benedict, 1902	4.0 (9.0)	0.3 (0.6)	
Miscellaneous crabs	Primarily anomurans, including lithode and hermit crabs; several brachyuran genera including Cancer, Pugettia, Scyra, and Oregonia	21.7 (35.0)	16.3 (23.5)	
Miscellaneous shrimps	Primarily Pandalus danae Stimpson, 1857, but also other members of the genus Pandalus, as well as the genera Lebbeus, Eualus, Heptocarpus and possibly others	37.0 (38.6)	15.8 (11.2)	
Giant Pacific octopus	Enteroctopus dofleini Wülker, 1910	0.1 (0.2)	0.0	
Cup corals	Balanophyllia elegans Verrill, 1864, Caryophyllia alaskensis Vaughan, 1941	6.7 (15.8)	22.1 (19.0)	
Benthic fishes	
Grunt sculpin	Rhamphocottus richardsonii Günther, 1974	0.1 (0.2)	0.1 (0.2)	
Longfin sculpin	Jordania zonope Starks, 1895	0.2 (0.4)	2.7 (4.1)	
Sailfin sculpin	Nautichthys oculofasciatus Girard, 1858	0.1 (0.2)	0.0	
Scalyhead sculpin	Artedius harringtoni Starks, 1896	0.8 (1.6)	1.8 (2.3)	
Miscellaneous sculpins	Cottid genera including Artedius, Orthanopias, Oligocottus, Radulinus, Chitonotus and possibly others.	5.5 (4.6)	0.7 (1.2)	

At each site we surveyed four transects (25 m long by 4 m wide) at depths between 8 and 15 m (chart datum). We quantified fish and invertebrate abundance by counting all individuals of the target taxa occurring fully or partly within 0.25 m2 quadrats placed at 15 random positions along each transect. We also estimated visually the percent cover of kelp (mainly the genera Agarum, Costaria, Laminaria and Saccharina) within the same quadrats.

Citizen-contributed (REEF) surveys

To verify that the patterns of echinoderm abundance detected at our 20 study sites reflected local trends accurately, we compiled qualitative data on the abundance of sunflower star and green urchin in Howe Sound and adjacent Indian Arm, east of Vancouver, from the Reef Environmental Education Foundation (REEF; REEF, 2014) citizen science database. Through REEF, scuba divers are trained in species identification and collect data on abundance of species sighted during recreational dives. Divers assign an abundance score from 1–4 to each species they can positively identify: score 1 = a single individual, 2 = 2–10 individuals, 3 = 11–100 individuals and 4 = >100 individuals. Species with no abundance score were assumed to be absent, which we deemed to be a fair assumption given that our target taxa were easy to identify.

We extracted the abundance scores of sunflower stars and green sea urchins for all REEF surveys submitted between January 1, 2010 and November 1, 2014 in Washington and BC. To depict trends in abundance over time, we plotted 60-day running averages of the abundance scores for both species. Missing values were filled in using linear interpolation.

Data analyses

We used linear mixed-effects models in the R statistical platform (nlme package; Pinheiro et al., 2015) to compare sunflower star abundance, green urchin abundance and kelp cover before and after the sea star mortality. We obtained sunflower and green urchin abundance for each transect by summing the number of sunflower stars and, separately, green urchins across all quadrats and log-transforming the values prior to analysis. Kelp cover was averaged across all quadrats within each transect. In all cases, we included ‘site’ as a random effect, and verified the assumptions of normally distributed residuals, homoscedasticity and the absence of leverage by visually examining quantile, residual vs. fitted and Cook’s distance diagnostic plots, respectively.

To depict graphically site-level changes in the abundance of sunflower stars, green sea urchins and kelp, we plotted the relative difference in abundance for each group at each site. Relative abundance was calculated as the abundance after the mortality event minus the abundance prior to it divided by the mean abundance for both time periods. Abundance was calculated as the total count of each species at each site for sunflower stars and green urchins, and as the average percent cover at each site for algae.

To compare overall benthic community composition before and after the sea star mortality, we ran a permutation-based, non-parametric analysis of similarity (ANOSIM; Clarke, 1993) using PRIMER (v. 1.0.3; Clarke & Gorley, 2006). Abundance matrices (species by site) were compiled for each period (i.e., pre- and post-mortality), in which abundance was estimated as the total count of each taxon across transects and/or quadrats at each site. The raw data were square-root-transformed to reduce the influence of very abundant or very rare species. Bray-Curtis similarity coefficients were computed between pairs of sites (Clarke & Warwick, 2001). The ANOSIM procedure was carried out on the similarity matrix. ANOSIM generates an R statistic, which varies between 0 (samples are as similar across groups as they are within group) and 1 (all samples within groups are more similar to each other than to any sample across groups) and is tested for difference from zero with a permutation test (in this study, N = 999 permutations). The differences in benthic assemblages were visualized in a non-metric, multidimensional scaling (MDS) plot in which samples that are more similar in community composition appear closer together than more dissimilar samples. Stress values of <0.1 suggest that distances among samples in an MDS plot accurately reflect the extent of community differences (Clarke & Warwick, 2001). Finally, we conducted an analysis of similarity percentages (SIMPER) to identify the main taxa responsible for any differences observed between pre- and post-mortality assemblages. We considered a taxon to be important to community differences if its individual contribution was 11% or more, which is twice the expected value if dissimilarity contributions were evenly distributed among all taxa in the analysis (i.e.,100 percent divided by 18 taxa, multiplied by 2). The SIMPER analysis also includes an indication of evenness, expressed as a consistency ratio (CR). CR is the average dissimilarity contribution of a taxon divided by the standard deviation in dissimilarity values of that taxon, for each time period. CR values greater than one suggest that the taxon contributed to dissimilarity between time periods equally across all sites (Terlizzi et al., 2005).

Results

Sea star mortality

At our monitored sites, the abundance of sunflower stars declined by 89% ± 29% (mean ± SD), from an average of 0.42 (±0.76) sunflower stars per m2 before the mortality event to 0.06 (±0.22) individuals per m2 after it (LME: t = 4.62, df = 139, p < 0.0001; Fig. 3). Three sites had no sunflower stars in 2009/2010, and were not included in the percent decline calculation. All 17 sites with sunflower stars in 2009/2010 had fewer sunflower stars in 2014.

Figure 3 Mortality of sea stars, and subsequent change in urchin abundance and kelp cover after sea star mortality.

Mean abundance (per m2) of (A) sunflower stars and (B) green sea urchins, and (C) percent cover of kelp on rocky reefs in Howe Sound, British Columbia, on 80 transects before and after the mass mortality of sea stars in 2013. Error bars represent standard error. The dominant kelp was the sea colander kelp, Agarum fimbriatum.

The REEF data included 1,568 surveys carried out at 28 sites broadly distributed across BC and Washington between 2010 and 2014. Although sunflower stars were sighted on 98% of surveys in the years before the mortality event and on 89% of surveys in the years afterward, a marked decline was evident in their abundance score trajectory (Fig. 4). At this larger geographic scale, sunflower stars started declining in approximately the third week of September, some 15 weeks after the first report of sea star wasting in the region.

We were unable to detect a geographic pattern in the spread of the sea stars’ mortality in our study area because of the speed at which the sea star wasting progressed. It was first observed in Howe Sound (at Whytecliff Park; 49°22′18.4″N, 123°17′33.8″W) on 2 September, 2013, and we then noted it at all of our study sites the following month.

Benthic community composition

There was a significant shift in overall community composition following sea star mortality in Howe Sound (ANOSIM: R = 0.326, p = 0.001; Fig. 5), and many species changed in abundance from one period to the next (Table 1). The community shift was largely driven by an increase in abundance of green urchins (Table 2). Green urchin abundance quadrupled after the near-disappearance of sunflower stars (LME: t = − 3.10, df = 139, p = 0.0023; Fig. 3). This trend is supported by the REEF surveys, although these qualitative data suggest that green urchin numbers began increasing in the first week of September, two to three weeks before the detectable onset of sea star decline (Fig. 4). There was also an increase in the abundance of cup corals, while the numbers of small shrimps and crabs decreased (Table 2). Cumulatively, these four taxa accounted for nearly two-thirds (62%) of the dissimilarity in benthic community composition before and after the sea star mortality, and their contributions were consistent across sites (CRs > 1; Table 2). Despite their marked decline, sunflower stars did not contribute disproportionately to the dissimilarity between time periods (SIMPER; individual contribution to dissimilarity = 7.15%). Overall, within-year similarity was higher after than before sea stars died (SIMPER; average inter-site similarity before = 46.28%, after = 58.11%; Fig. 5), suggesting that communities became more homogeneous following the sea star mortality.

Table 2 Differences in pre- and post-mortality benthic assemblages.

The four taxa that contributed disproportionately to dissimilarity in benthic community composition on rocky reefs before and after the 2013 sea star mass mortality, as well as the focal sea star, Pycnopodia helianthoides. Mean densities (# per 30 m2 ± 1 SD), consistency ratios, and individual and cumulative contributions (in %) to differences between years are shown. The consistency ratio is calculated as a species’ average dissimilarity contribution divided by the standard deviation of dissimilarity values. A consistency ratio > 1 indicates an even contribution to community dissimilarity across sites. The analysis was conducted on square-root-transformed data (see Methods) but untransformed densities are presented here.

Taxon	Mean density (SD)	Consistency ratio	Individual contribution	Cumulative contribution	
	Before	After		(%)	(%)	
Strongylocentrotus droebachiensis	18.3 (41.0)	77.2 (157.5)	1.09	18.91	18.91	
Cup corals	6.7 (15.8)	22.2 (19.1)	1.41	13.04	31.95	
Misc. shrimps	37.0 (38.7)	15.9 (11.2)	1.3	11.29	43.23	
Misc. crabs	21.7 (35.0)	16.3 (23.5)	1.05	11.15	54.38	
Pycnopodia helianthoides	6.4 (11.4)	0.9 (3.3)	1.18	7.15	69.05	

In addition to shifts in benthic animal community composition, there was also a change in the abundance of kelp. Kelp cover decreased from 4% (±10%) in 2009/2010 to <1% (±2%) in 2014 (LME: t = 2.669, df = 139, p = 0.0085; Fig. 3). In all years, the kelp at our sites was almost exclusively the sea colander kelp, Agarum fimbriatum, but also included Saccharina latissima.

Figure 4 Sunflower star and green sea urchin abundance trajectories.

Sixty-day running average abundance scores for green sea urchins (Strongylocentrotus droebachiensis; green solid line) and sunflower stars (Pycnopodia helianthoides; purple dashed line) recorded in REEF surveys from January 2010 to November 2014 in Washington and British Columbia (n = 1568 surveys). Grey bands indicate 95% confidence intervals of the running average. The vertical red dotted line indicates the date of the first recorded observation of sea star wasting syndrome (7 June 2013), which was on the Olympic coast of WA.

Figure 5 Rocky reef species assemblages before and after sea star mortality.

Multidimensional scaling plot of benthic community composition on rocky reefs before (blue triangles) and after (inverted red triangles) the 2013 sea star mass mortality event in Howe Sound, British Columbia. The analysis included 18 fish and invertebrate taxa at 20 sites, surveyed both in 2009/2010 and 2014. The associated stress value (0.13) suggests some distortion in the multivariate representation of the data.

Figure 6 Relative difference in abundance of sea stars, urchins and kelp by site.

The relative difference in total count of sunflower stars (blue triangles) and green urchins (green circles), and the relative difference in the mean percent cover of algae (red squares) before and after the sea star mass mortality. Open symbols indicate sites where population density was zero both before and after the mass mortality. Relative difference was calculated as the change in abundance divided by the mean abundance of both time periods. A relative difference of −2 indicates the population declined to zero. Sites are numbered chronologically according to the order in which they were surveyed, from June to August, 2014. The geographic location of these sites is shown in Fig. 2.

At the regional scale, the changes in abundance of sunflower stars (decline), green urchins (increase) and kelp (decline) were consistent with a trophic cascade (Fig. 3). At the site level, the patterns were more variable (Fig. 6). Eleven of the 17 sites that had some P. helianthoides before the sea star mortality showed increases in green urchin abundance concomitant with declines in sea star abundance (Fig. 6). Eight of these 17 sites showed declines in kelp cover concomitant with increases in green urchin abundance (Fig. 6). A clear alternation of population trajectories from predators to herbivores to kelp was clear at eight of the 16 sites (Fig. 6).

Discussion

The wasting disease that affected echinoderms in the northeast Pacific in 2013/2014 heavily impacted populations of sunflower stars, the sea stars that formerly dominated subtidal communities. We found a noticeable shift in benthic community structure following the sea star decline. Community changes were largely driven by changes in the abundance of green sea urchins, cup corals, shrimps and crabs. The temporal coincidence of the alternating trajectories of abundance of sea stars, urchins and kelp, as well as the overlapping distributions and documented trophic linkages among these three taxa, meet the diagnostic criteria of a tri-trophic cascade (Grubs et al., 2016), triggered by the mass mortality of predatory sunflower stars.

Sunflower star densities declined by almost 90%, on average, at our sites in Howe Sound, BC. Such a decline in sea stars rivals the largest magnitudes reported for disease-induced echinoderm mass mortalities (e.g., 70% of Strongylocentrotus droebachiensis in Nova Scotia Scheibling & Stephenson, 1984; 95% of S. franciscanusin California Pearse et al., 1977; 97% of Diadema antillarum across the Caribbean Lessios, 1988). The percent change in biomass of P. helianthoides must be even greater than the change in relative abundance because the sea stars we observed following the mortality event were almost exclusively juveniles (<6 cm diameter). The very large individuals (>50 cm diameter) present before the mortality event likely played a larger role in structuring benthic communities than the juveniles present after the event. The steep decline in sunflower star numbers, occurring some 15 weeks after the first official sighting of sea star wasting, was clearly evident in the qualitative density scores generated by citizen science (REEF) surveys, which covered a broader geographic area. The time-series of REEF data suggests that sunflower star population levels were somewhat variable, perhaps reflecting variation in the sites surveyed by divers, but largely stable between the first snapshot in 2009/2010 and the onset of the wasting event in 2013. More generally, the benthic species composition of the Strait of Georgia region has remained remarkably stable in recent decades, even in the face of climate regime shifts (Marliave et al., 2011). It therefore seems unlikely that the sea star population declines, and concomitant changes in benthic community composition, could be ascribed to a different, unreported disturbance occurring prior to 2013.

The most striking change we observed in community composition was a marked increase in the abundance of green urchins. Overall, green urchins were nearly four times more numerous following the sea star mortality event than before. However, the mechanism of this population increase remains unclear. One possibility is that a recruitment pulse of green urchins coincided with sea star wasting disease, which would have generated a large urchin cohort even in the presence of sunflower stars. Another possibility is that urchin recruits—whether part of a normal or a large cohort—were able to survive better in the absence of abundant sea star predators (Duggins, 1981). The size (3–5 cm diameter) of the majority of urchins present a year following the sea star mortality makes this explanation perhaps unlikely. Green urchins of this size on the east coast of North America are at least three years of age, and possibly more than a decade old (Russel, Ebert & Petraitis, 1998; Vadas et al., 2000). If these growth rates are similar on the Pacific coast, then most of the urchins we saw could have settled several years before the sea star mortality event. However, urchin growth rates can be highly variable (Vadas et al., 2000), depending on food supply and temperature (Thompson, 1983; Meidel & Scheibling, 1999; Pearce et al., 2005), Urchin growth rates have not yet been estimated in BC. A third possible explanation is that the observed increase in urchin abundance resulted from a shift in urchin behaviour following the sea star mortality event. The impact of ‘intimidation’ on predator–prey interactions can be as important as direct consumption (Lima & Dill, 1990; Preisser, Bolnick & Benard, 2005). Under risk of predation, prey individuals alter a suite of behaviours, including habitat choice, foraging range, and time under cover (Werner et al., 1983; Peacor & Werner, 2001; Trussell, Ewanchuk & Bertness, 2003; Schmitz, Krivan & Ovadia, 2004). The effect of sunflower stars on urchin behaviour is well documented. In field experiments in Alaska, both green and purple (S. purpuratus) urchins moved away after P. helianthoides arms were placed in the centre of urchin aggregations (Duggins, 1981), and urchin distribution shifted rapidly when sea star abundance was experimentally increased (Duggins, 1983). Fear-released urchins could therefore respond by moving from refuges, perhaps in very shallow or deep habitats or in sheltered crevices inaccessible to sea stars (and divers), to more open substrates, making them easier to see and count.

The data from REEF surveys support a behavioural rather than a consumptive mechanism for the increase in urchin numbers. Whereas one would expect a delayed increase in urchin numbers following a release from predation (Wangersky & Cunningham, 1957), green urchin numbers began to increase at approximately the same time as the decline in sunflower stars was evident (Fig. 4). The observed change in green urchin abundance may therefore be due, at least in part, to green urchins modifying their distribution in response to the decline of sunflower stars.

Another conspicuous change we observed was a ∼80% reduction in kelp cover (Fig. 3), pointing to a potential trophic cascade triggered by the sea star mortality event. There are many documented examples of urchin abundance directly influencing the abundance of algae (e.g., Fletcher, 1987; Carpenter, 1990; Estes & Duggins, 1995; McClanahan et al., 1996; Palacin et al., 1998; Scheibling, Hennigar & Balch, 1999; Villouta et al., 2001). As urchin numbers rise, either due to a large recruitment event (Hart & Scheibling, 1988) or the absence of a predator (Watson & Estes, 2011), kelp is rapidly depleted. The alternating directions of population trends of sea stars, urchins and kelp observed here are consistent with the hypothesis of a trophic cascade triggered by the sea star disease. The tri-trophic cascade was clearly evident at the larger scale of Howe Sound (Fig. 3), but detectable at only half of the sites, with a few additional sites showing only part of the cascade (Fig. 6). It is notable that the sites surveyed earliest (i.e., sites 1–5 on Figs. 2 and 6) showed an increase in kelp cover, perhaps because not enough time had passed for changes to take place. At other sites where the trophic cascade was not detectable, it is possible that urchins moved elsewhere in search of better food sources (e.g., at sites 17 and 18 on Fig. 6), or that the presence of juvenile sea stars (i.e., site 20 on Fig. 6) resulted in different trophic interactions.

In contrast to green urchins, the abundance of many prey species did not increase in the near-absence of sunflower star predators. For example, there was no change in the abundance of red urchins (S. franciscanus) and white urchins (S. pallidus). Neither species is common in Howe Sound, and little is known about the ecology of S. pallidus. However, S. franciscanus may generally be less susceptible to sea star predation than other urchin species because they grow too large to be consumed (Duggins, 1981). Moreover, although crustaceans constitute a significant portion of the diet of sunflower stars (Shivji et al., 1983; Estes & Duggins, 1995; Lambert, 2000), shrimps and crabs declined following the sea star mortality. Several of the crustaceans we monitored use kelp for both food and habitat. The spot prawn, Pandalus platyceros, for instance, specifically uses sea colander kelp as nursery habitat (Marliave & Roth, 1995). The decline of some crustacean taxa could result from the reduced kelp cover and therefore be a fourth step in the cascade documented here.

Another fourth link in the ecological cascade triggered by sea star mortality might involve cup corals. Their increase in abundance was surprising as cnidarians are not normally consumed by P. helianthoides (Shivji et al., 1983; Herrlinger, 1983). However, cup corals are known to fare poorly in areas dominated by macroalgae (Fadlallah, 1983). Contact with algae causes coral polyp retraction, which in turn allows overgrowth by filamentous and coralline algae (Coyer et al., 1993). Increases in density of cup corals can be swift (<1 year), and of the magnitude observed here (3–4 times), after algae disappear (Coyer et al., 1993). Of course, the reduced abundance of kelp and of sea stars may also have allowed for a less obstructed view of the substrate by the observers. As a number of taxa were not monitored in this study, there were likely other changes following the sea star mortality event that we did not detect.

In conclusion, our study contributes to understanding the ecological consequences of the northeast Pacific sea star mass mortality. The most notable change was a marked increase in the number of green sea urchins, which might have already had trickle-down effects on other levels of the ecosystem by the time we detected it. It is unclear whether the changes observed will persist as long-term consequences of the near-disappearance of sea stars. Nonetheless, further monitoring will help elucidate the resilience of this ecosystem in the face of acute biological disturbances. Although such a sudden and drastic decline in sea star populations is alarming, it provides a large-scale natural experiment that may advance our understanding of subtidal trophic cascades and invertebrate population dynamics.

Supplemental Information

Data S1 Data from transects and behavioural experiment

Click here for additional data file.

Supplemental Information 1 R script for data analysis and figures

Click here for additional data file.

We thank the Vancouver Aquarium Marine Science Centre for all aspects of this study, and in particular the Howe Sound Research team: Jeff Marliave, Donna Gibbs, Laura Borden, and Boaz Hung. Thank you to volunteer divers Roya Esragh, Marielle Wilson, Brian Caron, Justin Lisaingo, Crystal Kulstar, and Alex Clegg. Comments by Jeff Marliave, Alejandro Frid, Jane Williamson and one anonymous reviewer greatly improved the manuscript.

Additional Information and Declarations

Competing Interests

Author Contributions

Data Availability

The authors declare there are no competing interests.

Jessica A. Schultz conceived and designed the experiments, performed the experiments, analyzed the data, wrote the paper, prepared figures and/or tables, reviewed drafts of the paper.

Ryan N. Cloutier performed the experiments, wrote the paper, reviewed drafts of the paper.

Isabelle M. Côté conceived and designed the experiments, wrote the paper, reviewed drafts of the paper.

The following information was supplied regarding data availability:

The raw data has been supplied as Data S1 and the code has been supplied as a Supplemental Information 1.

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
