# Peer review of "Evidence for a trophic cascade on rocky reefs following sea star mass mortality in British Columbia"

_PeerJ, doi:10.7717/peerj.1980_

## Round 0.1 · original submission · Minor Revisions

Both reviewers were positive about your MS, but made a number of constructive suggestions that need addressing. As reviewer 2 says, it would be useful if you could show site-level changes in relative abundances of starfish, urchins (& ideally kelp). This could potentially be done using the REEF data by overlaying different-coloured lines for each of the three taxa on a (very) small graph for each of the 28 monitoring sites, so the reader can see how consistently the trophic cascade occurs. The graphs could be ordered geographically if the starfish dieoff occurred in a wave. Also, can you speculate as to why abundances of red & white urchins didn't increase following the starfish die-off?

Reviewer 1 ·

Basic reporting

The authors have done an adequate job of this but improvements have been suggested leow.

Experimental design

Below I suggest that the authors couch their work in terms of testing hypotheses.

Validity of the findings

The findings are valid but some suggestions are made below. Some excessive speculation in the discussion which should at least be identified.

Additional comments

This paper presents some interesting data about community structure on British Columbia reefs following the starfish mass wasting disease. The paper is fairly well written and I think is acceptable for publication with some revision. The major suggestion I have is this: why do the authors not present their results in terms of questions and hypotheses rather than ad-hoc-sounding comparisons? Surely the authors realize that we have ample information regarding what these starfish eat. My guess is that the authors wanted to look at whether sea star decreases led to increases in their prey, and concomitant trophic cascades given that the main prey are urchins. The paper would be much stronger if written to reflect that, rather than the present MS, which comes across as pretending that the authors had basically no idea that Pycnopodia eats urchins and had to do an experiment to support that. I suggest that the authors clearly state the hypotheses they had going into the study up front in the introduction and proceed from there, altering the discussion to reflect the support or lack thereof for each hypothesis, rather than speculating on why each species might have been affected by the dieoff event.

Some more specific comments follow.

Line 46: “decimation” technically means reduced by 10%.

Line 58: This taxonomic terminology including the name of the describer (Brandt here) is distracting and unnecessary. If it’s not a specific journal formatting requirement I suggest removing it.

Line 59-60: “Several” is an understatement!

Line 68-69: I would not put this statement in if pers comm is all that can be done to support it.

Lines 86-88: I’m not sure if I would say this. There have been many people collecting data on this since day one, and even if this statement turns out to be true in terms of first published study (and it might not end up that way) it is unnecessarily divisive in my opinion.

Line 115: This scale is not logarithmic, even roughly.

Lines 124-137: This experiment is interesting but many like this have been done before including with the same two species. E.g. see the following paper, which the authors cite but do not summarize.

Duggins, D. O. (1981). Interspecific facilitation in a guild of benthic marine herbivores. Oecologia, 48(2), 157-163.

Line 145: what tests were used to evaluate the assumptions?

Line 210-211: Given that kelp cover was measured by visual estimation, do the authors really think they have two decimal places of precision in these estimates?

Line 237-238: “suggests?” We already know that Pycnopodia eats urchins, including this species.

Lines 252-254: It is likely that an analysis of this kind would show a difference in community composition between any two periods regardless of the existence of a recognized “disturbance” like this (and as an aside I think this stretches the definition of the term disturbance).

Line 280-281: As do other studies.

Line 287: Any data or even idea on what those refuges are? Are the authors talking about movement out of cracks etc into the open, or a larger scale migration from shallower depths for example?

Line 299: Is there any evidence for this “smothering” effect (which is a vague description)?

Lines 299-308: all ad-hoc speculation.

Fig. 2: No error bars on this graph? I think this might look better as a bar graph.

Fig 3: If possible confidence intervals should be indicated on these graphs. I am sure that the number of surveys must vary quite a bit over time and this would contribute to variation. It appears that the reef surveys recorded the dieoff before the official “first noticed” data. If true that would be an interesting thing to point out.

Fig 4 is strange looking, looks like an optical illusion drawing. I suggest limiting the number of lines (axes) to three.

Line 360: missing space

Fig 5: Is this significantly different? Normally I prefer boxplots but the authors might consider a bar graph here.

Fig 6: Greater spacing between bar pairs would help with the aesthetics of this plot.

Table 3 results could be included in the text of the Results.

·

Basic reporting

This manuscript assesses the impact of a mass sea star mortality on community structure in British Columbia. It is a well written piece which cites appropriate literature. My only comment on literature cited is that it is a little parochial in content and the authors could also include citations from other parts of the world to strengthen some of their more generic statements on sea urchins and algae. The structure of the manuscript fits well into the templates used by PeerJ and all figures and tables are well presented and relevant to the manuscript. I would, however, like to see some form of error bars on Figure 3 (please see further comments below) and a scale bar on Figure 1. In general, the manuscript is of a quality appropriate for PeerJ and conforms to PeerJ's guidelines. I have not assessed the availability of raw data due to my current location and access during this review, and will rely on the other reviewers and the editor to check.

Experimental design

The manuscript has clearly defined questions and hypotheses that are relevant and meaningful, and is original research that provides pertinent information on echinoderm diseases and community structure. Ethically, the manuscript conforms to standards in this field of research.
It appears that this is one of those fortuitous times when a longer-term monitoring program really comes into its own. The authors have had several years of data on community structure in the area prior to any observations of sea star wasting and are thus able to show quite string relationships in abundance in organisms once wasting was observed.
What is missing in this manuscript, however, is any comment or assessment of where wasting was first observed and how it spread throughout Howe Sound. This is of particular importance as the authors discuss a concurrent increase in sea urchin abundance. This may be the case but is impossible to tell as the data are presented. I suggest that the authors show some type of site-specific analysis or visual presentation on the decrease of sea stars and increase of sea urchins, and incorporate some discussion on where the disease was first observed and the rate of spread in the area. Images of a sea star with wasting disease versus a more 'normal' looking specimen could also help here.

Validity of the findings

Findings within the manuscript are, in general, valid but I caution the authors not to extrapolate too far from their findings. For example, the behavioural experiment shows that sea urchins will move away from sea stars at a greater rate than an inanimate object. From these results the authors then state (p13, line 280) that the sea urchins were eliciting 'strong avoidance behaviour'. Once again, this may be the case but the authors would need to include some positive controls into their design to make this statement. It is also difficult for the authors to talk about community shifts (as the title suggests) when they acknowledge that "a number of taxa were not monitored in this study' (p 14, line 307).
Conclusions directly relate to the original question and the discussion on explanations of the sea star decline and sea urchin abundance interesting.

Additional comments

While I have recommended minor revisions here, I urge you to reanalyse these data in a more site-specific manner to really nail the relationship between your wasting sea star and the increase in sea urchin abundance. An interesting study!

---

## Round 0.2 · Minor Revisions

Note from Staff: Apologies for this delayed decision. The Editor did make a decision (see the text below) a while back, but there was a technical glitch which meant it was not sent at the time. Our apologies for this inadvertent additional delay.

Editor decision text:

Dear Jessica, thank you for resubmitting the MS - you've made a nice job of the revision. I'm happy to accept it once you make the following very minor edits:
Line 44: space needed: “pathogen(Lessios et al., 1984)”
Lines 94, 218, 297: “top-down” is redundant here: “top-down trophic cascade”
Line 191: change to “sea stars’ “, i.e., add an apostrophe
Line 192: This sentence needs rewriting: “Sea star wasting progressed so rapidly that by the time it was first observed in Howe Sound (at Whytecliff Park; 49°22'18.4"N, 123°17'33.8"W) on 2 September, 2013, it was soon present at all sites that were subsequently investigated.”
Line 276: change to “sunflower stars”
Line 325: change to “for a less”
Fig. 4 is good but I found the units on the X-axis a bit confusing. Was using a log response ratio or simply expressing the final value as a proportion of the initial not possible due to zero abundance values in the denominator? If you keep the current units you need to at least state in the legend that “-2” means a decline to zero.

---

## Round 0.3 · accepted · Accept

Thank you for making the required changes.